# Integrated exome and RNA sequencing of dedifferentiated liposarcoma

Makoto Hirata ⓘ et al.[#]

The genomic characteristics of dedifferentiated liposarcoma (DDLPS) that are associated with clinical features remain to be identified. Here, we conduct integrated whole exome and RNA sequencing analysis in 115 DDLPS tumors and perform comparative genomic analysis of well-differentiated and dedifferentiated components from eight DDLPS samples. Several somatic copy-number alterations (SCNAs), including the gain of 12q15, are identified as frequent genomic alterations. *CTDSP1/2-DNM3OS* fusion genes are identified in a subset of DDLPS tumors. Based on the association of SCNAs with clinical features, the DDLPS tumors are clustered into three groups. This clustering can predict the clinical outcome independently. The comparative analysis between well-differentiated and dedifferentiated components identify two categories of genomic alterations: shared alterations, associated with tumorigenesis, and dedifferentiated-specific alterations, associated with malignant transformation. This large-scale genomic analysis reveals the mechanisms underlying the development and progression of DDLPS and provides insights that could contribute to the refinement of DDLPS management.

[#]A full list of authors and their affiliations appears at the end of the paper.

Dedifferentiated liposarcoma (DDLPS) is a rare malignant tumor with an incidence of <0.1/million each year[1,2] that occurs in ~10% of cases of intermediate (locally aggressive) well-differentiated liposarcoma (WDLPS)[3]. Surgical excision is the primary treatment modality used for DDLPS, as DDLPS exhibits a low response rate to conventional chemotherapeutic reagents[4]. To date, several analyses have unveiled genomic characteristics common to DDLPS, including the amplification of 12q13-15, that are also frequently found in WDLPS[5–10]. These studies have also identified a number of genes within 12q13-15, including *HOXC13*, *MDM2*, *HMGA2*, *CDK4*, and *CPM*, as being key to the development of DDLPS and WDLPS; a number of additional genomic occurrences, such as the loss of 11q23 and the gain of 6q23 and 1p32, have been defined as genomic abnormalities that are specific to DDLPS[6,8,11–17]. Recently, the Cancer Genome Atlas (TCGA) Research Network has identified the characteristics of some types of soft-tissue sarcoma, including DDLPS with amplification of 12q13-15, through comprehensive genomic analysis and showed that the classification of DDLPS tumors based on the status of their somatic copy-number alterations (SCNA) and DNA methylation could predict clinical prognosis[18]. These results, based on the analysis of 50 DDLPS cases, still require validation. In addition, the genomic events associated with the malignant transformation of DDLPS, and with DDLPS tumors without 12q13-15 amplifications that are histologically diagnosed, remain to be identified.

We established the Japan Sarcoma Genome Consortium (JSGC) in 2014 with the aim of generating a comprehensive map of the genomic alterations and abnormalities present in bone and soft-tissue tumors, in order to facilitate the implementation of precision medicine. Here, we collect tumor and normal tissue samples from 65 patients with DDLPS and perform whole-exome and RNA sequencing at two facilities, the Institute of Medical Science at the University of Tokyo (hereinafter, JSGC-IMSUT), and the National Cancer Center Research Institute, Japan (JSGC-NCC). In addition, we obtain FASTQ data derived from the whole-exome and RNA sequencing of 50 DDLPS tumors from TCGA, in order to conduct genomic meta-analysis on a total of 115 patients with DDLPS. In addition, eight pairs of well-differentiated (WD) and dedifferentiated (DD) components from DDLPS tumors are obtained for the comparison of their genomic alterations.

## Results

**Clinical characteristics of the subjects.** A total of 115 patients were enrolled in the current study (28 from JSGC-IMSUT, 37 from JSCG-NCC, and 50 from TCGA), and clinical information was collected from 108 of 115 patients. Of these 108 patients, 75 (69.4%) were male, and the mean age at diagnosis was 62.7 (±12.7) years (Table 1). A total of 73.1% of the DDLPS tumors arose from the retroperitoneum or abdomen, while the distribution of the primary tumor sites varied among the three groups; tumors were most frequently located in an extremity in JSGC-IMSUT patients (66.7%) and in the retroperitoneum or abdomen in JSGC-NCC (78.4%) and in TCGA (86.0%) patients. A total of 76.4% of the patients had tumors that were 10 cm or more in diameter (Table 1; Supplementary Table 1) and 98.1% of the patients underwent surgery (Table 1; Supplementary Table 1). These results indicated that the patients enrolled in this study may be representative of the general population of patients with DDLPS.

**Somatic mutations and copy-number alterations.** Based on exome sequencing for 115 pairs of DDLPS and normal tissue samples, we identified 2639 somatic mutations, including

### Table 1 Clinical characteristics of patients with DDLPS.

| Features | Total (n = 108)[a] |
|---|---|
| Male sex, n (%) | 75 (69.4) |
| Age at diagnosis ± std (y) | 62.7 ± 12.7 |
| Primary site, n (%) | |
| Retroperitoneum or abdomen | 79 (73.1) |
| Extremity | 25 (23.1) |
| Chest wall or back | 4 (3.7) |
| Tumor size, n (%) | |
| 10 cm ≥ | 25 (23.6) |
| 10 cm < | 81 (76.4) |
| Unknown | 2 (—) |
| Local treatment, n (%) | |
| Surgery only | 97 (89.8) |
| Surgery with adjuvant radiation | 9 (8.3) |
| Radiation | 1 (0.9) |
| Heavy ion | 1 (0.9) |
| Surgical margin, n (%) | |
| R0 | 40 (37.7) |
| R1 | 58 (54.7) |
| R2 | 3 (2.8) |
| RX | 5 (4.9) |
| Not applicable | 2 (—) |

[a]Clinical information from 108 of 115 patients was available for the current study

nonsynonymous single nucleotide variants (SNVs) and short insertions/deletions (INDELs), with a mean of 24.2 (0.274 per coding megabase) and a range of 0 to 70 mutations (Fig. 1a) per sample. The frequency distribution of the somatic mutations was almost comparable among the three groups, JSGC-IMSUT, JSGC-NCC and TCGA (Supplementary Fig. 1a). The mutation frequency at each chromosome ranged from 0.114 (in chromosome 21) to 0.482 (in chromosome 12) per coding megabase (Fig. 1b; Supplementary Fig. 1b). Base substitution analysis of synonymous and nonsynonymous SNVs with the adjacent 5′ and 3′ flanking nucleotides, in order to better understand the mutational processes involved, showed that nucleotide alterations from anyCG to anyTG were the most frequently detected in DDLPS (Fig. 1c). The nucleotide alterations exhibited similar trends among the three groups (Supplementary Fig. 1c). Mutation signature analysis, using the COSMIC database, showed that signature 3 contributed the most to these base substitutions, followed by signature 1 (Fig. 1d), indicating that both the failure of DNA double-strand break-repair (signature 3) and aging (signature 1) may contribute to the development of DDLPS. Recurrently, mutated genes (frequency of more than five samples) were *MUC16*, *TTN*, *ATRX*, *TRHDE*, *PCLO*, *ZNF717*, *TP53*, *FLG*, and *NAV3*; however, the mutated loci at each gene were not recurrent (Supplementary Fig. 2). The GISTIC analysis of SCNAs identified 28 gained regions (357 genes) and 55 lost regions (455 genes) (Fig. 1e; Supplementary Data 1 and 2). As expected, the gain of 12q15 resulted in the lowest FDR q-value in the GISTIC analysis. The genome-wide analysis of SCNAs and their corresponding GISTIC values showed that the copy numbers of genes in chromosomes 11 and 13, and the short-arm of chromosome 9 were generally decreased, while those of the long-arms of chromosomes 9 and 20 and the short-arms of chromosomes 4, 5, 7, 19, and X were increased. Notably, the copy numbers of the genes located at 12q14.1–15 greatly increased (Supplementary Fig. 3). Among these genes, *SLC35E3*, *MDM2*, and *CPM* exhibited the highest mean GISTIC values with the lowest standard deviations (Supplementary Fig. 3, boxed area), indicating that these three genes are greatly and consistently amplified in DDLPS.

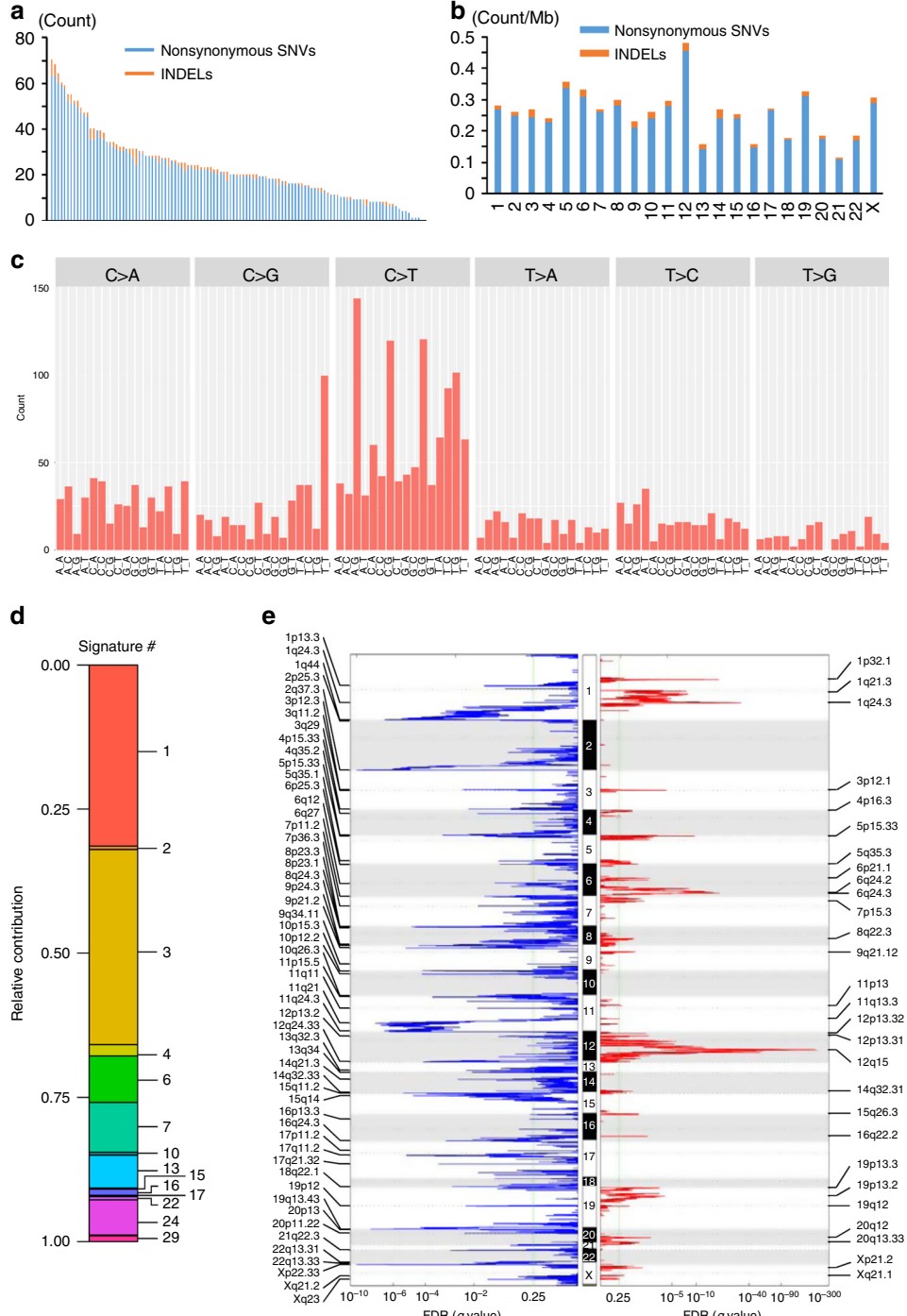

**Fig. 1** Characteristics of the somatic mutations and copy-number alterations in DDLPS. **a** Frequency of nonsynonymous SNVs and short INDELs identified by exome sequencing for each DDLPS sample. **b** Mean mutation frequency per megabase of coding sequence for each autosomal chromosome. Light blue and orange bars represent the frequency of SNVs and short INDELs, respectively. **c** 96 substitution classification for DDLPS samples. SNVs were classified according to six base substitution patterns, C > A, C > G, C > T, T > A, T > C, and T > G, and also based on the identity of the bases immediately 5′ and 3′ to each mutated base. **d** Mutation signature analysis for 119 DDLPS samples. The values represent the contribution of each signature (left) and the signature number (right). **e** Chromosomal regions with gained (red) and lost (blue) SCNAs identified in 119 DDLPS samples using GISTIC 2.0. The genes in each region are listed in Supplementary Data 1 and 2.

**Recurrent chromosomal rearrangements and fusion genes.** Genomon-Fusion, based on the sequencing of RNA from 101 DDLPS samples, revealed that the long-arm of chromosome 12 was the most frequent site of intra-and interchromosomal rearrangements, followed by chromosome 1 (Fig. 2a; Supplementary Fig. 4a). Most of the interchromosomal rearrangements occurred between

chromosome 12 and other chromosomes (Fig. 2a), and the frequency of interchromosomal rearrangements between chromosome 12 and other chromosomes was significantly correlated with that of intra-chromosomal rearrangements within the other chromosomes ($r = 0.937$ and $P = 4.97 \times 10^{-11}$, Pearson's test), indicating that instability within chromosome 12 was associated with the frequency

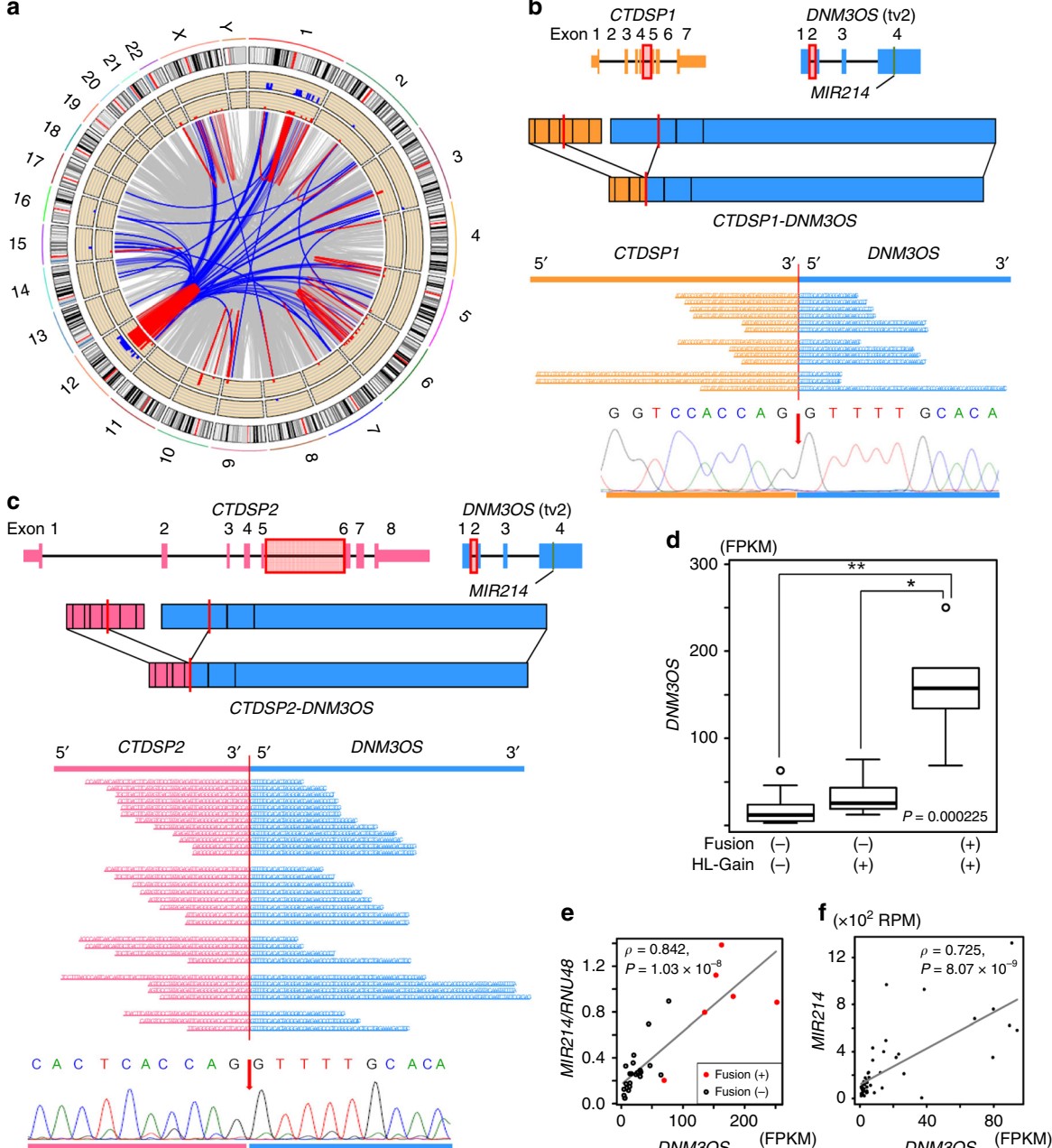

**Fig. 2** Chromosomal rearrangements and fusion genes in DDLPS. **a** Circos plot of chromosomal rearrangements across 103 DDLPS tumors. The central circle displays structural rearrangements of the fusion genes. The red and blue lines indicate intra- and interchromosomal rearrangements, respectively, which were recurrently observed in at least six cases, while the gray lines show those which were observed in less than six cases. Case count at each gene was performed by referring to the number of partner-genes; a gene rearranged with multiple positions in one case was recognized as different cases. The second and third inner circles represent the histograms of the cases with intra- and interchromosomal rearrangements, respectively, of the genes at the indicated positions. The range of the axis for the counts of the cases in each histogram is from 0 to 60. **b**, **c** Schematic of CTDSP1-DNM3OS (**b**) and CTDSP2-DNM3OS- (**c**) fusion genes. The region in the red box includes the genomic breakpoint, and the red vertical bars denote the breakpoints in the mRNA (upper panel). Supporting reads (middle) and sequencing chromatogram (lower) of CTDSP1-DNM3OS from three DDLPS tumors. The red arrow represents the breakpoint. **d** DNM3OS expression in DDLPS. DDLPS tumors from the JSGC-NCC cohort were classified according to their DNM3OS-fusion status and the high-level copy-number gain (HL-Gain) of DNM3OS. The box signifies the upper and lower quartiles; the center bold line within the box, median; the upper and lower whiskers, upper and lower quartiles $+/-$ interquartile ranges, respectively. *$P < 0.05$ and **$P < 0.01$ by Steel-Dwass test. **e**, **f** Pearson correlation tests and scatter plots showing the relationship between the expression of MIR214 and DNM3OS in JSGC-NCC (**e**) and TCGA (**f**) tumors. The expression of MIR214 and DNM3OS in 30 and 52 DDLPS tumors from JSGC-NCC and TCGA, respectively, was analyzed. The red dots represent DNM3OS-fusion-positive samples. P-values, derived from Pearson's rank correlation test.

of whole chromosomal rearrangements in DDLPS. Genomon-Fusion identified three recurrently occurring interchromosomal fusion genes: C15orf7-CBX3, CTDSP1-DNM3OS, and CTDSP2-DNM3OS (Fig. 2b, c; Supplementary Table 2). As the presence of

C15orf7-CBX3 as a germline mutation has been previously reported[19], we focused on the CTDSP2-DNM3OS and CTDSP1-DNM3OS (CTDSP1/2-DNM3OS) fusions, which were verified by capillary sequencing of cDNA from the tumor samples (Fig. 2b, c). We further

identified several other *DNM3OS* fusions that had the same break-point, including *CTDSP1/2-DNM3OS* and *CPM-DNM3OS*, in eight of 101 DDLPS samples (Supplementary Table 3). *DNM3OS* transcripts were significantly increased in *DNM3OS*-fusion-positive DDLPS (Fig. 2d). A gene set enrichment analysis (GSEA) of both *DNM3OS*-fusion-positive and -negative samples also showed that the upregulation of *DNM3OS* was correlated with the presence of a high GSEA score (Supplementary Fig. 5a) and the significant enrichment of cell-cycle-related gene sets, including those associated with the G2M checkpoint, E2F targets, and mitotic spindles, in positive samples (Supplementary Fig. 5b), indicating the association of *DNM3OS*-fusion genes with cell-cycle regulation. *DNM3OS* encodes the *MIR199A2-MIR214* cluster[20,21]; as the expression of *MIR214* has been found to mirror that of *DNM3OS* during embryonic development[22], we further analyzed the correlation between *DNM3OS* and *MIR214*. The *DNM3OS*-fusion genes found in the DDLPS specimens still contained *MIR214* even after their genomic translocation (Fig. 2b, c). Scatter plots of the expression of *DNM3OS* and *MIR214* in JSGC-NCC and TCGA samples showed significant correlation in the levels of these transcripts (Fig. 2e, f), indicating that *DNM3OS* translocation may cause the induction of *MIR214* expression. Neither additional 17 WDLPS samples nor 8 well-differentiated components, obtained from the DDLPS tumor samples, were found to harbor the *DNM3OS* fusions; however, other interchromosomal fusion genes between chromosomes 1 and 12 were detected in nine of the 17 WDLPS cases (Supplementary Fig. 4b), suggesting the possibility that *DNM3OS* fusions might serve as specific DDLPS markers to discriminate WDLPS cases with and without dedifferentiation potential.

**Association of SCNAs with clinical outcomes**. Among the genomic alterations that were identified in DDLPS during a series of next-generation sequencing experiments, the alterations that reoccurred with more than 10% frequency were subjected to analysis of their associations with clinical outcomes. Log-rank and univariate Cox-regression analyses revealed that 12 SCNA regions, including those involving the gain of 1p32.1, 4p16.3, 5p15.33, 6p21.1, 20q12, Xp21.2, and Xq21.2, and the loss of 2q37.3, 9p21.2, 9q34.11, 13q34, and 16q24.3, were identified as significant predictors of poor progression-free survival (Supplementary Fig. 6a and Supplementary Table 4a). Further multivariate Cox-regression analyses that included these 12 SCNAs identified the high-level gain of 4p16.3 and 6p21.1 and the loss of 9q34.11 as a significant independent predictor of poor progression-free survival (Supplementary Table 4a). In contrast, 15 SCNAs, including those involving the gain of 1p32.1, 5p15.33, 5q35.3, 19p13.3, 19q12, and Xq21.1 and the loss of 6q27, 9p24.3, 9p21.2, 9q34.11, 11p15.5, 11q24.3, 13q32.3, 13q34, and 18q22.1, were significantly associated with the disease-specific survival of patients with DDLPS (Supplementary Fig. 6b and Supplementary Table 4b); the gain of 1p32.1 was independently associated with poor disease-specific survival (Supplementary Table 4b).

**Mutational landscape of dedifferentiated liposarcoma**. The mutational profiles and genomic alterations that were found to be associated with DDLPS are summarized in Fig. 3a. As the TCGA study included DDLPS cases that were defined by 12q13-15 amplifications[18], the cluster analysis for DDLPS with the high-level gain of 12q15 was conducted after the classification of DDLPS without the high-level gain of 12q15 as Cluster 3. One SCNA, that involving the gain of 1p32.1, was independently associated with disease-specific survival and was the basis for dividing the DDLPS cases into two major clusters: Cluster 1 harbored the high-level gain of 12q15 along with the gain of 1p32.1, while Cluster 2 showed only the gain of 12q15 (Fig. 3a). Histological examinations, including immunohistochemistry and

FISH, verified the compatibility of Cluster 3 samples with DDLPS. Among the recurrently mutated genes, mutations or the copy-number loss of *TP53* was found to be accumulated in Cluster 3, particularly in three of the five DDLPS samples that did not have the high-level gain of 12q15 (Fig. 3a), indicating that the disruption of the *MDM2/TP53* axis was the most decisive genomic event contributing to DDLPS development. Copy-number analysis also identified the common genomic features of Cluster 3, including the consistent gain or loss of 157 genes in nine regions (Supplementary Data 3), some of which were associated with PI3K-AKT signaling based on the KEGG pathway database.

**Association of genomic alterations with clinical prognosis**. Survival analysis after genomic clustering showed favorable progression-free survival rate in patients with Cluster 2 DDLPS compared with patients with Cluster 1 using Kaplan–Meier and univariate Cox-regression analyses (Fig. 3b and Table 2a). The disease-specific survival in patients with Cluster 2 was also more favorable than in patients with Cluster 1 (Fig. 3c and Table 2b). Multivariate Cox-regression analyses showed that Cluster 1 classification (vs Cluster 2) was a significant predictor for poor progression-free and disease-specific survival, independently of the surgical margin and primary tumor site (Table 2a, b). Further multivariate analysis including the SCNA regions independently associated with progression-free survival (i.e., the high-level gain of 4p16.3 and 6p21.1, gain of Xq21.1 and loss of 9q34.11) and significant clinical parameters also demonstrated that these four SCNA regions are independent predictors of poor progression-free survival (Supplementary Table 5). As GISTIC analysis identified 83 significant SCNA regions (Fig. 1e), we further explored the possibility that additional SCNAs in Cluster 1 or 2 could have affected the clinical prognosis. Within Cluster 1, log-rank tests showed that nine gained regions (4p16.3, 6p21.1, 6q24.2, 6q24.3, 11q13.3, 19p13.3, 20q12, 20q13.33, and Xq21.1) and three lost regions (11q24.3, 13q32.3, and 13q34) were significantly associated with progression-free survival (Supplementary Fig. 7a and Supplementary Table 6a); it was also demonstrated that six gained regions (5q35.3, 14q32.31, 19p13.2, 19p13.3, 20q12, and 20q13.33) and seven lost regions (5q35.1, 6q27, 8q24.3, 10p15.3, 11q24.3, 13q32.3, and 13q34) were significantly associated with disease-specific survival (Supplementary Fig. 7b and Supplementary Table 6b), A multivariate Cox-regression analysis that included these regions identified the loss of 13q32.3 as an independent predictor of progression-free survival. In contrast, an analysis of Cluster 2 DDLPS showed that the additional high-level gain of 1q24.3, 4p16.3, 6p21.1, and Xq21.1, and the loss of 9q34.11, 10p15.3, 12q24.33, and Xq22.33 were associated with poor progression-free survival (Supplementary Fig. 8a and Supplementary Table 7a) and that the additional gain of 1q24.3, 5p15.33, 12p13.32, 19q12, Xp21.2, and Xq21.1 and the additional loss of 1q24.3, 6q27, 9q34.11, and 18q22.1 were associated with poor disease-specific survival (Supplementary Fig. 8b and Supplementary Table 7b). Further multivariate Cox-regression analysis of Cluster 2 showed that the high-level gain of 1q24.3, 4p16.3, and Xq21.1, and the loss of 9q34.11, 12q24.33, and Xp22.33 were independent predictors of progression-free survival (Supplementary Table 7). To further explore the SCNA-dependent alterations of gene expression that were associated with clinical prognosis, we examined the transcript levels of genes in the SCNA regions. Six genes, *JUN*, *DNM3*, *DNM3OS*, *TAF9B*, *DGKQ*, and *STX18*, which are located at 1p32.1, 1q24.3, Xq21.1, and 4p16.3, showed altered expression that was correlated with the SCNAs in all three cohorts (Supplementary Table 8) and some of the genes exhibited significant association of high-expression with poor clinical prognosis (Supplementary Fig. 9);

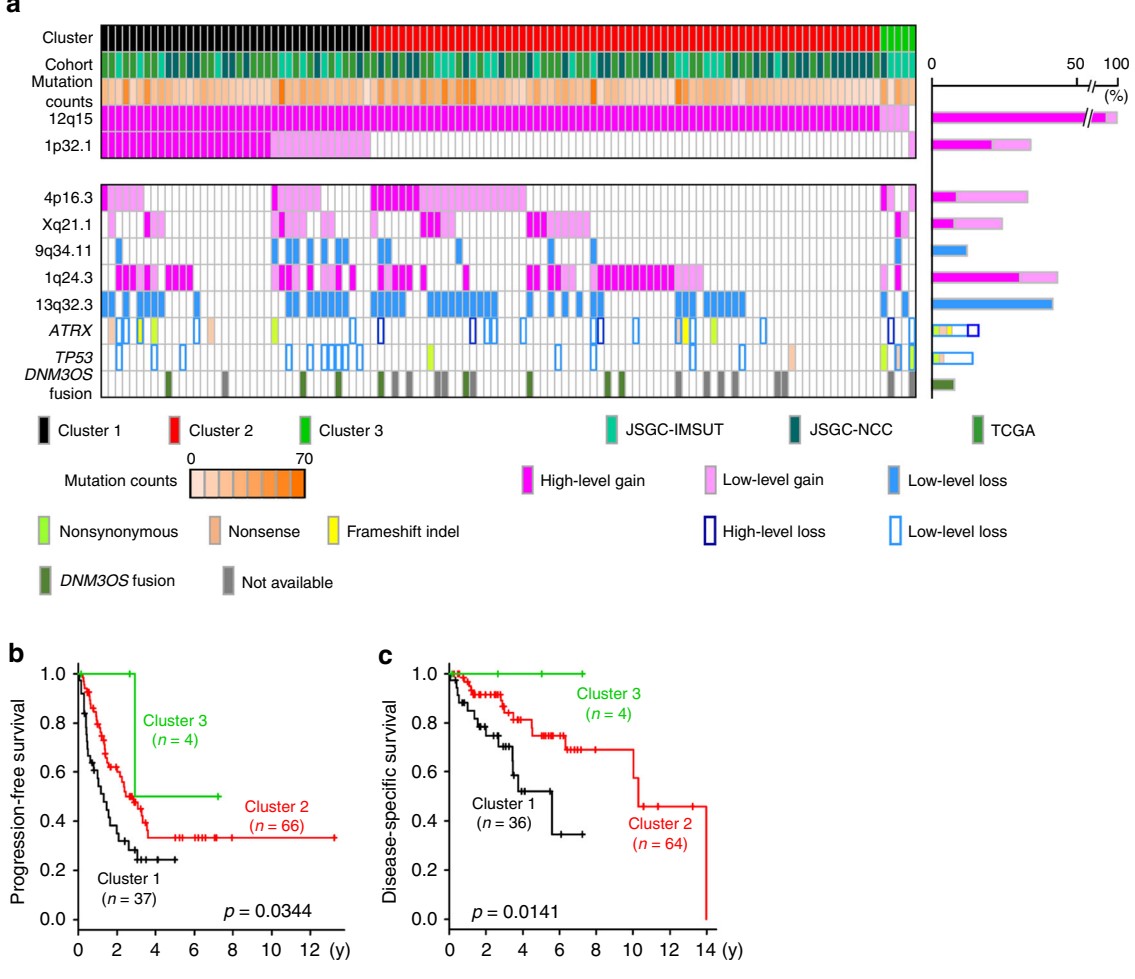

**Fig. 3** Mutational landscape of DDLPS and survival analysis by genomic clustering. **a** Mutational landscape of DDLPS. The status of the SCNAs, which were independently associated with disease-specific or progression-free survival, *TP53* and *ATRX* driver mutations, and *DNM3OS*-fusion genes is depicted in the landscape. Clustering was performed based on the SCNA status of 1p32.1 and 12p13.32 in DDLPS with a high-level gain of 12q15. The bar graph on the right side represents the ratio of the affected samples to all examined samples. **b**, **c** Impact of genomic clustering on disease-specific (**b**) and progression-free (**c**) survival of patients with DDLPS. Disease-specific and progression-free survival was analyzed using Kaplan–Meier methods. *P*-values derived from log-rank analysis are included on each panel.

this indicated that these genes that contained SCNAs were the initial drivers of tumor progression in DDLPS.

**Comparative analysis between WD and DD components**. As we obtained both intermediate well-differentiated (WD) and high-grade malignant dedifferentiated (DD) components from eight DDLPS samples, we compared their genomic profiles in order to determine the mechanisms underlying the malignant transformation of DD. As expected, DD harbored more somatic mutations than matched WD in all cases, but shared few somatic mutations with WD (Fig. 4a; Supplementary Fig. 10a). In contrast, WD and DD shared more SCNA regions in common, while DD harbored more prominent SCNAs as well as additional SCNA regions when compared with WD (Fig. 4b; Supplementary Fig. 10b). GISTIC analysis confirmed the results of the comparative SCNA analysis, and identified the shared gain of 1q24.3 and 12q14.3-15 and loss of 1p36.33, 15q11.2, and 16p13.3 between DD and WD (Supplementary Fig. 10c, d). Circos plots showed common recurrent intra- and interchromosomal rearrangements at chromosomes 1 and 12 in the DD and WD pairs (Fig. 4c; Supplementary Fig. 11a), while the heatmap of the chromosomal rearrangements revealed that the frequency of

rearrangements was increased in DD compared to WD (Supplementary Fig. 11b, c). These results indicated that SCNAs and chromosomal rearrangements at chromosomes 1 and 12, but not somatic mutations, were common initial genomic events in both DD and WD and that additional copy-number alterations or chromosomal rearrangements were associated with the development of DD.

The multidimensional scaling of the RNA expression from six matched pairs revealed a clustered expression profile for WD but a relatively scattered expression profile for DD (Fig. 4d). GSEA revealed that gene sets that were related to cell-cycle progression, including G2M checkpoint and E2F targets, were significantly enriched in DD, while those related to adipocyte differentiation or lipid metabolism, including adipogenesis and fatty acid metabolism, were enriched in WD (Fig. 4e, f; Supplementary Table 9a, b). We finally performed genome-wide screening analysis to identify genes that are involved in the malignant transformation of DD. During the first screening step, we searched for genes with recurrent SCNAs that are specifically found in DD and identified 133 gained genes in 20 regions and 305 lost genes in 37 regions (Supplementary Fig. 12). In the second step, we examined the expression levels of the 438 genes in DD and WD, and identified 27 genes that showed differential expression in accordance with

**Table 2 Cox-regression analysis of progression-free (a) and disease-specific (b) survival with genomic clustering.**

| | Univariate | | | Multivariate | | |
|---|---|---|---|---|---|---|
| | HR | (95% CI) | *P*-value | HR | (95% CI) | *P*-value |
| *(a) Progression-free survival* | | | | | | |
| Primary tumor site | | | | | | |
| Trunk (vs Extremity)[a] | 5.22 | (2.08–13.10) | $4.29 \times 10^{-4}$** | 4.29 | (1.59–11.58) | $4.01 \times 10^{-3}$** |
| Surgical margin | | | | | | |
| (R2, R1, R0) | 2.25 | (1.33–3.81) | $2.52 \times 10^{-3}$** | 1.62 | (0.87–3.01) | 0.131 |
| Genomic cluster | | | | | | |
| Cluster 1 (vs 2) | 1.82 | (1.08–3.04) | 0.0234* | 2.31 | (1.33–4.00) | $2.84 \times 10^{-3}$** |
| Cluster 3 (vs 2) | 0.399 | (0.55–2.92) | 0.365 | 1.12 | (0.14–8.85) | 0.912 |
| *(b) Disease-specific survival* | | | | | | |
| Primary tumor site | | | | | | |
| Trunk (vs Extremity)[a] | 8.14 | (1.10–60.06) | 0.0397* | 5.93 | (0.76–46.21) | 0.0894 |
| Surgical margin | | | | | | |
| (R2, R1, R0) | 2.70 | (1.16–6.26) | 0.0207* | 2.23 | (0.84–5.86) | 0.106 |
| Genomic cluster | | | | | | |
| Cluster 1 (vs 2) | 2.86 | (1.28–6.39) | 0.0104* | 3.18 | (1.35–7.48) | $8.07 \times 10^{-3}$** |
| Cluster 3 (vs 2) | $1.38 \times 10^{-8}$ | (0–Inf) | 0.999 | $1.55 \times 10^{-7}$ | (0–Inf) | 0.998 |

*CI* confidence interval, *HR* hazard ratio
The results are presented for the univariate and multivariate Cox-regression analysis for progression-free and disease-specific survival, using clinical measures and genomic cluster. Kaplan–Meier survival curves according to the SCNA regions are shown in Fig. 3. *$P < 0.05$, **$P < 0.01$
[a]Trunk includes abdomen, retroperitoneum, chest wall, and back, and extremity includes extremity, shoulder, and girdle

the SCNAs (Fig. 4g, h; Supplementary Table 10a, b). Of note, the expression levels of *G0S2* and *DGAT2* were remarkably decreased in DD compared with WD (Supplementary Table 10b), suggesting the involvement of the downregulation of *G0S2* and *DGAT2* in malignant transformation mediated by copy-number loss in DDLPS.

## Discussion

This study examined the genomic alterations associated with DDLPS by conducting whole-exome and RNA sequencing of more than 100 tumor samples. Through a series of analyses, we confirmed that the gain of the chromosomal region 12q15, which is already well-known to be associated with DDLPS, is the most frequent mutation observed in DDLPS; we also identified a number of *DNM3OS*-fusion genes. Based on the status of the genomic alterations, DDLPS could be classified into three groups, and this genomic classification could predict clinical outcomes. In addition, the comparative analysis of WD and DD revealed the SCNAs and chromosomal rearrangements at chromosomes 1 and 12 to be common initial genomic events and also revealed that the augmentation of the initially gained SCNA regions, the occurrence of additional SCNA regions, and/or further chromosomal rearrangements are events that are specifically associated with DD.

Previous studies have repeatedly reported the copy-number gain at 12q13-15 in DDLPS[5–10]. Most of these studies focused on the fact that the copy-number gain of specific genes, including *MDM2*, *HMGA2*, and *CDK4*, were driver genomic alterations[7]. The current study also identified the gain of 12q15 as the most frequent event in DDLPS (Fig. 1e) and notably distinguished *MDM2*, *CPM*, and *SLC35E3* as the most consistently and greatly duplicated genes in this region (Supplementary Fig. 3), indicating that the simultaneous gain of *MDM2*, *CPM*, and *SLC35E3* is a crucial step during the development of DDLPS. This study found that five of the 115 DDLPS tumors harbored no or low-level amplification of 12q15. Indeed, it is difficult to diagnose malignant soft-tissue tumors with little or no gain of 12q15 as DDLPS, but the histological diagnosis of the JSGC patients in Cluster 3 was verified by musculoskeletal pathologists both before and after the genomic analysis. The evidence that somatic mutations or the copy-number loss of *TP53* were accumulated in Cluster 3 (Fig. 3a) could support the histological diagnosis and suggests the

necessity of the disruption of the *MDM2-TP53* axis during the development of DDLPS.

Some histological types of sarcoma can be characterized by the presence of specific fusion genes, such as *EWSR1-FLI* in Ewing's sarcoma[23], *EWSR1-ATF1* or *EWSR1-CREB1* in clear cell sarcoma[24], and *SS18-SSX1/2* in synovial sarcoma[25], all of which function as drivers of tumor development. *FUS-DDIT3* is frequently associated with myxoid liposarcoma[26], while no recurrent fusions have been reported in DDLPS. The current study identified *CTDSP1-DNM3OS* and *CTDSP2-DNM3OS* as recurrent fusion genes. As *CTDSP1* and *CTDSP2* encode the C-terminal domain small phosphatases 1 and 2 and the knock-down of *CTDSP2* in DDLPS cell lines inhibited cell proliferation[27], further analysis is essential to characterize the expression and function of the fusion protein. However, and more importantly, DDLPS that contained *DNM3OS*-fusion genes showed the significant upregulation of *DNM3OS* (Fig. 2d; Supplementary Fig. 5a) and were correlated with cell-cycle pathways when compared with those without fusion genes (Supplementary Fig. 5b). In addition, the gain of 1q24.3, accompanied by the upregulation of *DNM3* and *DNM3OS* (Supplementary Table 8), was significantly associated with poor progression-free survival in Cluster 2 DDLPS (Supplementary Table 7). Because *DNM3OS* encodes the *MIR199A2-MIR214* cluster[20,21], of which *MIR214* was maintained in the *DNM3OS*-fusion genes (Fig. 2b, c), the fusion genes may be involved with the induction of *MIR214*. Indeed, *MIR214* and *DNM3OS* are consistently expressed during embryonic development[22], and their expression was highly correlated in DDLPS (Fig. 2e, f). Several tumor-suppressor genes, including *PTEN*, *ATM*, *TP53*, and the adipogenic transcription factor *PPARD*, have been validated as targets of *MIR214*[20,28]. Taken together, these lines of evidence suggest that the upregulation of *DNM3OS* mediated by chromosomal rearrangement could lead to the proliferation of tumor cells and contribute to DDLPS progression.

Based on a series of genomic analyses, the DDLPS tumors could be classified into three groups (Fig. 3a). This classification showed that Cluster 1 DDLPS tumors were associated with poorer clinical outcomes than Cluster 2 DDLPS tumors (Fig. 3b, c and Table 2). Cluster 1 was characterized by the gain of 1p32.1, which contains *JUN* and other genes, and is comparable with K1 and a portion of the K2 clusters from the TCGA study[18]. The

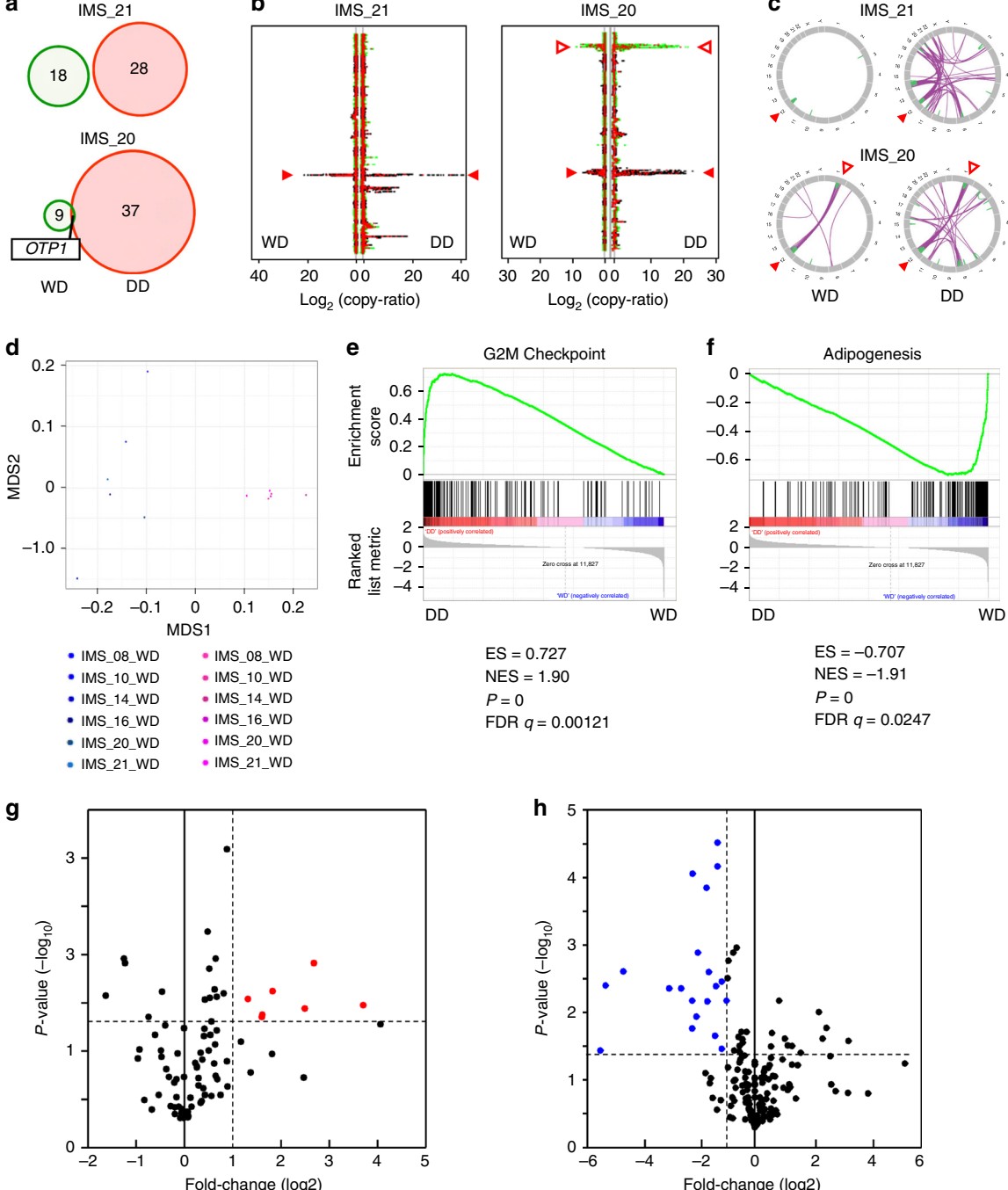

**Fig. 4** Comparative analysis of well-differentiated (WD) and dedifferentiated (DD) components from DDLPS. **a–c** Representative somatic mutations (nonsynonymous SNVs and INDELs) (**a**), SCNAs (**b**), and SVs (**c**) in WD and DD components from the same patients. In **a**, the circle size and numbers indicate the number of somatic mutations in WD or DD. A boxed gene, *OTP1*, indicates a common somatic mutation in the sample. In **b**, copy numbers were plotted according to the order of the chromosomal regions, from chromosome 1 (top) to 22 (bottom) and chromosome X. Red lines indicate segmented exome circular binary segmentation calls. The segmentation size is based on the exome capture kit bed file. Solid arrows indicate 12q15; empty arrows indicate 1p32.1. In **c**, two representative cases are presented, with others presented in Supplementary Fig. 9. **d** Multidimensional scaling analysis of expression profiles of paired WD and DD. Six pairs of WD and DD were analyzed. **e, f** GSEA analysis comparing the expression profiles of DD and WD. The gene sets most enriched in DD (**e**) and WD (**f**) are shown. **g, h** Volcano plots of the DD-specific gain (**g**) and loss (**h**) of genes. The red and blue dots denote large magnitude fold-changes (more than 2 or less than ½; horizontal axis) and high statistical significance (more than 1.301 of −log₁₀ of *P*-value by one-sided paired *T* test; vertical axis), respectively.

current and previous studies also showed the upregulation of *JUN* was correlated with the gain in its copy number (Supplementary Table 8)[18]. Because *JUN* amplification blocks adipogenesis[11] and is oncogenic in liposarcomas[13], the upregulation of *JUN* that occurs as a result of 1p32.1 gain may play a pivotal role in DDLPS

progression[18,29]. Prognostic nomograms, that provide survival predictions for sarcoma patients has been established[30,31]. Genomic clustering was shown to be an independent prognostic factor of the clinical parameters; primary tumor site and surgical margin (Table 2), and further multivariate Cox-regression analysis,

including majority of MSKCC-nomogram clinical parameters; age, gender, surgical margin, primary site, and tumor size[30], showed significant association of the genomic clustering with progression-free and disease-specific survivals (Supplementary Table 11). Such clinical models can more precisely predict prognosis by considering the relevant genomic clustering information in addition to the clinical parameters. The results of a previous CNA assay combined with a comparative genomic hybridization (CGH) array on 52 DDLPS samples showed that the loss of 11q23–24 was the most common mutational event in DDLPS, and that the loss of 19q13 was associated with poor prognosis[6]. The current study also identified 11q24.2 and 19q13.43 as recurrent SCNAs (frequency of 40.3% and 26.1%, respectively), but failed to show the association of 11q24.2 or 19q13.43 with clinical outcomes. We also identified the genes, whose expression level was dependently modulated on the prognostic SCNAs (Supplementary Table 8). These recurrent SCNAs and genes can be the potent prognostic marker, though further validation analysis, using other cohort samples, is required for the clinical application.

The comparative analysis of the genomic alterations in WD and DD from the same tumor tissue identified frequent SCNAs and chromosomal arrangements, especially in chromosomes 12 and 1, but few common somatic mutations (Fig. 4a–c; Supplementary Figs. 10, 11). This evidence supports the influence of two important factors that contribute to the mechanisms underlying the development of DDLPS. First, SCNAs, especially at 1q24.3 and 12q14.3-15, and concurrent inter- and intra-chromosomal rearrangements in chromosomes 1 and 12, but not somatic mutations, were initial occurrences that were shared during the development of these types of tumors. Second, the augmentation of the common SCNAs and the emergence of additional SCNAs and chromosomal rearrangements in chromosome 12, both of which did not occur in WD, may cause the malignant transformation of DD. Previous comparative analysis between DDLPS and WDLPS identified the loss of 11q23 and the amplification of 6q23 and 1p32 as genomic abnormalities specific to DDLPS[6,11]. This study identified these SCNAs as recurrently affected regions in both DD and WD. This discrepancy might be caused by the differing backgrounds of the WD components in DDLPS and WDLPS. Indeed, the current fusion analysis identified DNM3OS fusions only in DDLPS and not in WDLPS, and a previous microarray-based transcriptome analysis showed a distinct gene expression pattern in WD from DDLPS versus that from WDLPS[32]. Another previous CGH array that compared pairs of WD and DD components from DDLPS tumors failed to identify any SCNAs that were able to significantly distinguish the two types of components[33]. Further comparative analysis using next-generation sequencing may discriminate the genomic profiles of WD in DDLPS from those of WDLPS and provide important information for use in establishing a treatment strategy for these tumors.

This study identified 27 genes that were specifically gained or lost in DD, but not in WD, and were differentially expressed in accordance with the alteration in their copy number (Fig. 4g, h; Supplementary Table 10a, b). Of the 27 genes, G0S2 (G0/G1 Switch 2) and DGAT2 (diacylglycerol O-acyltransferase 2) were highly expressed in WD, with a mean FPKM of ~1200 and 70, respectively, but were remarkably suppressed in DD at a level of approximately 95% (Supplementary Table 10b). As G0S2 regulates lipid metabolism and promotes apoptosis by binding to BCL2[34–36], and DGAT2 plays an important role in triacylglycerol biosynthesis and fat digestion and absorption[37], the copy-number loss and concomitant downregulation of G0S2 and DGAT2 most likely strongly induced the dedifferentiation and malignant transformation of adipogenic cells.

Overall, the genomic alterations that occur during the progression of DDLPS can be summarized as follows (Fig. 5): The common genomic alterations in the DD and WD components, including the gain of 12q15 (containing MDM2, CPM, and SLC35E3), arise during the initiation step of DDLPS and lead to the impairment of P53 and chromosomal instability. During the second step of the malignant transformation, some of the tumor clones undergo further augmentation of the initial SCNAs and gain additional SCNAs and chromosomal rearrangements, including the DNM3OS-fusion genes and the loss of G0S2 and DGAT2, which contribute to the cell-cycle progression and impairment of adipogenesis. Finally, additional SCNAs, including the gain of 1p32.1, 1q24.3, 4p16.3, and Xq21.1 and the loss of 9q34.11, 12q24.33, 13q32.3, and Xp22.33, occur, which were found to be involved in tumor progression and are associated with poor clinical outcomes.

In conclusion, this study revealed the genomic characteristics of DDLPS using comprehensive genomic analysis of more than 100 tumor samples and revealed the genomic clustering of DDLPS tumors, which can be used to predict the prognosis of DDLPS patients. These findings will shed light on the underlying mechanisms of DDLPS development and progression and provide insights that can contribute to the refinement of DDLPS therapy.

## Methods

**Patients and tumor samples.** We collected matched pairs of frozen normal and tumor samples from 65 patients (28 in JSGC-IMSUT and 37 in JSGC-NCC) with dedifferentiated liposarcoma (DDLPS). We also obtained WD components from 8 of the 65 DDLPS samples. We used blood, skin, or adipose tissue as the germline control samples. All of the samples collected from JSGC-IMSUT were transferred to a core analytic facility after anonymization at each hospital. Other samples that were collected from JSGC-NCC were prepared for next-generation sequencing at the National Cancer Center Research Institute. The frozen tumor samples from JSGC-IMSUT were sectioned for histological evaluation (Supplementary Fig. 13) and extraction of DNA and RNA. Histological data from the frozen and formalin-fixed paraffin-embedded tumor samples, which had been prepared for clinical diagnosis, were evaluated by musculoskeletal pathologists to confirm the diagnosis and validity of the tumors and also to examine the content of the tumor cells. The present protocols were reviewed and approved by the Ethics Committees of all participating institutions, including the Institute of Medical Science, the University of Tokyo, the National Cancer Center, Japan, Tokyo Metropolitan Cancer and Infectious Diseases Center Komagome Hospital, Kyushu University, Osaka International Cancer Institute, Chiba Cancer Center, Nagoya University Graduate School of Medicine, Kanagawa Cancer Center, National Hospital Organization Hokkaido Cancer Center, and RIKEN Center for Integrative Medical Sciences. All of the participants were enrolled and anonymised after approval by the institutional review board. We obtained written informed consent from all participants, except for those who we could not contact due to loss of follow-up or death at registration. In these cases, the Institutional Review Boards at each participating institution granted permission for existing tissue samples to be used for research purposes. In addition, the Institutional Review Board of the Institute of Medical Science, University of Tokyo provided permission for the fully anonymised genetic data to be shared (protocol numbers 26-22-0630 and 30-78-B0305). None of the samples used in this study came from patients who had opted out of participation.

**Whole-exome sequencing.** The whole-exome sequencing of the 65 DD and 8 WD components as well as 65 matched germline samples was performed using target capture with Agilent SureSelect XT Human All Exon V5 + lncRNA (Agilent, 5190-6448) in JSGC-IMSUT and with Agilent SureSelect XT Human All Exon V5 (Agilent, 5190-6210) in JSGC-NCC. The raw sequence data generated by the Illumina HiSeq2000 or HiSeq2500 sequencers were processed through an in-house pipeline used for the whole-exome analysis of paired cancer genomes at the Human Genome Center, Institute of Medical Science, University of Tokyo.

We also obtained FASTQ sequence data for 50 DDLPS cases from TCGA, which were merged with our sequence data and subjected to the following analyses.

**Analysis of somatic mutations.** For our sequencing data, FASTQ files were generated by CASAVA 2.0. Candidate somatic mutations were identified using the Genomon pipeline [https://github.com/Genomon-Project/genomon-docs/tree/v2.0]. The human reference file that was used is GRCh37/hg19. The candidate mutations in a tumor sample were identified using the following criteria: (i) Fisher's exact $P \leq 0.01$; (ii) $\geq 5$ variant reads in the tumor sample; (iii) variant allele frequency (VAF) in the tumor sample $\geq 0.08$; and (iv) VAF of the matched normal

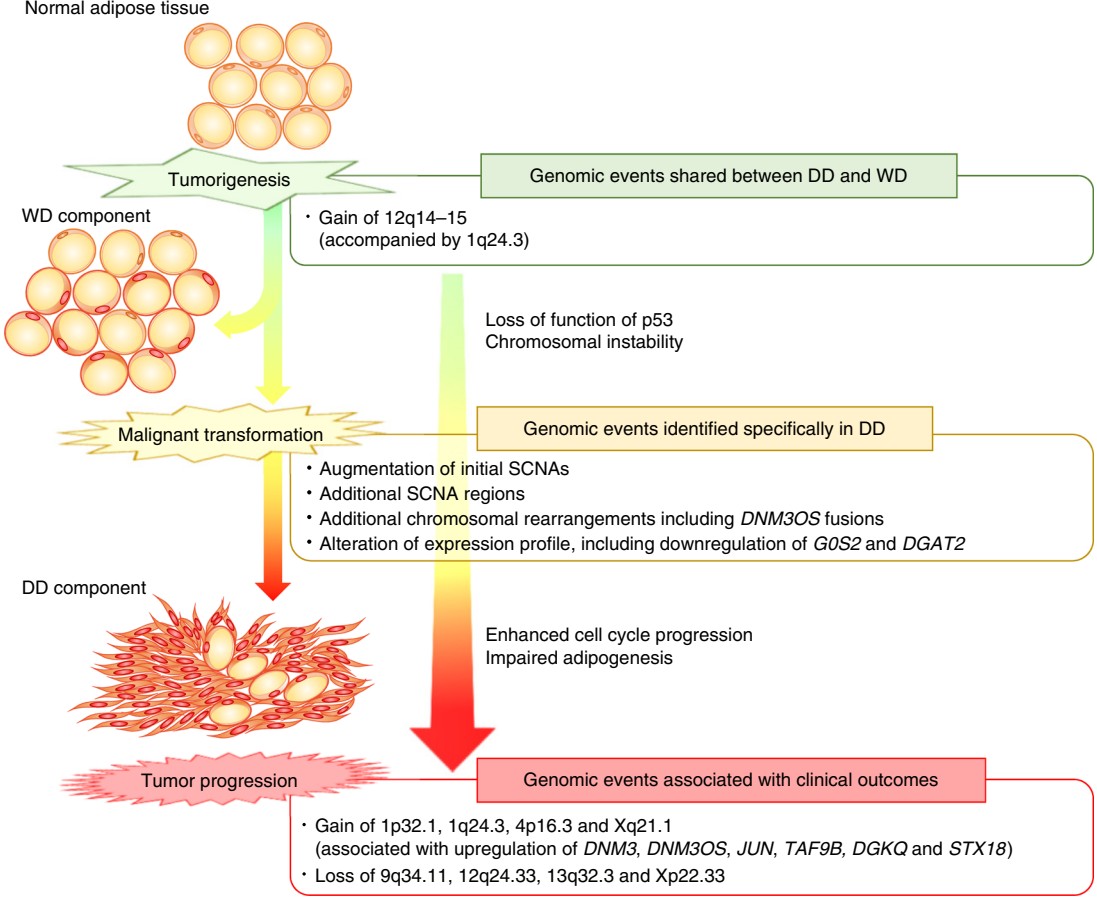

**Fig. 5** Scheme of genomic events during DDLPS progression.

sample < 0.07, with the exclusion of synonymous SNVs and known variants listed in NCBI dbSNP build 131.

**Analysis of somatic copy-number alterations**. Copy-number aberrations were quantified and reported for each bed size as the segmented, normalized, log2-transformed exon coverage ratio between each tumor sample and its matched normal sample. Significant focal copy-number alterations were identified using GISTIC2 (v 2.0.22)[38] [http://www.mmnt.net/db/0/0/ftp-genome.wi.mit.edu/distribution/GISTIC2.0].

**Mutational signature**. We used MutationalPatterns[39] to compare our mutational catalog to the previously identified COSMIC mutational signatures[40] [https://cancer.sanger.ac.uk/cosmic/signatures_v2].

**RNA sequencing**. In JSGC-IMSUT, the total RNA was extracted from frozen tumor tissues using the Qiazol reagent (Qiagen) and was purified using a RNeasy Plus Universal Mini kit (Qiagen) with DNase I digestion, according to the manufacturer's instructions. The RNA integrity was verified using an Agilent 2100 Bioanalyzer with RNA Nano reagents (Agilent Technologies). High-quality RNA from 15 DD and 6 WD samples was subjected to polyA + selection and chemical fragmentation, and the 100-200-bp RNA fraction was used to construct cDNA libraries using the TruSeq Stranded mRNA Prep kit (Illumina) according to the manufacturer's protocol. For the RNA-seq of low-quality RNAs from 4 DD samples and one WD sample, libraries were constructed from the total RNA using the TruSeq RNA Access Library Prep kit (illumina), which captured the coding regions of the transcriptome. In JSGC-NCC, the total RNA was extracted from 32 DDLPS and 17 WDLPS samples using ISOGEN reagent (Nippon Gene), and was purified using an RNeasy MinElute Cleanup kit (Qiagen). Libraries were constructed from total RNA using the TruSeq Stranded Total RNA with Ribo-Zero Gold kit (illumina). These paired-end libraries were sequenced with the Illumina HiSeq2000 or HiSeq2500 platform.

**Analysis of fusion genes**. The fusion transcripts were detected using Genomon (ver.2.2 [https://github.com/Genomon-Project/genomon-docs/tree/v2.2.com/Genomon-Project/genomon-docs/tree/v2.2]) and further filtered by excluding candidates that (i) were mapped to repetitive regions, (ii) had <3 spanning reads,

(iii) occurred out of frame, or (iv) had junctions that were not located at known exon–intron boundaries.

**Analysis of gene expression**. Gene expression values were estimated from the RNA-seq data from the tumor samples using Tophat2[41] (Tophat 2 v2.1.0 [http://ccb.jhu.edu/software/tophat/downloads/tophat-2.1.0.tar.gz]) and Cufflinks[42] (cufflinks v. 2.2.1 [http://cole-trapnell-lab.github.io/cufflinks/releases/v2.2.1/]). The paired-end transcriptome sequencing reads were aligned to the human reference genome (GRCh37/hg19) in Tophat2. BAM files named accepted_hits.bam, which were generated by the Tophat mapping module, were used to quantify the expression data using Cufflinks. Each gene expression dataset, derived from a different RNA library kit, was analyzed separately, as the RNA library kits each produced different expression profile clusters. A GSEA to identify gene sets enriched with DNM3OS-fusion-positive samples or DD components was performed using the JAVA GSEA v3.0 program[43]. Quantitative RT-PCR was performed twice to examine the MIR214 expression level in 29 JSGC-NCC samples. Each experiment was conducted in triplicate and the MIR214 expression levels were calculated by normalization to RNU48. The mean of the results of two experimental was used for further correlation analysis.

**Statistical analysis for clinical variants**. We obtained clinical information, including sex, age at diagnosis, primary tumor site, tumor size, modality of local treatment, and surgical margin, from 112 of 119 participants (Table 1). The mean follow-up duration for the 112 patients with DDLPS was 3.61 years, with a total of 401 person-years. Clinical factors, including age at initial presentation, sex, tumor size (10 cm or more vs less than 10 cm), primary site (retroperitoneum, abdomen or chest wall vs extremity), surgical margin status, metastasis status at presentation, and genomic status were analyzed for their association with progression-free and overall survival using the Cox proportional hazards regression model and Kaplan–Meier statistics. Log-rank tests determined the univariate significance of a factor. Factors found to be significant in a univariate analysis were included in a multivariate Cox proportional hazards regression model. The hazard ratio (HR) and 95% CI were used to report the magnitude of the differences and the strength of the association.

**Reporting summary**. Further information on research design is available in the Nature Research Reporting Summary linked to this article.

## Data availability

Sequencing FASTQ data files from exome and RNA sequencing have been deposited at the Japanese Genotype-phenotype Archive (JGA), which is hosted by the DDBJ, under accession number JGAS00000000177 and JGAS00000000182. Other data sets referenced during the study are available from the Genomic Data Commons [https://gdc.cancer.gov/]. All the other data supporting the findings of this study are available within the article and its Supplementary Information files and from the corresponding author upon reasonable request. A reporting summary for this article is available as a Supplementary Information file.

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

## Acknowledgements

We express our gratitude to all the participants and collaborators in the Japan Sarcoma Genome Consortium. We thank Satoyo Oda and Akane Sei for their technical and administrative support. We also thank Dr. Nobuyuki Hashimoto and Dr. Nobuto Araki for supporting the sample collection in Osaka International Cancer Institute. This study was supported by KAKENHI (16H02676) of Japan Society of Promotion of Science; by the Project for Development of Innovative Research on Cancer Therapeutics (P-DIRECT) from Japan Agency for Medical Research and Development (AMED) (15cm0106141h0002 and 15cm0106142h0002); by the Project for Cancer Research and Therapeutic Evolution (P-CREATE) from AMED (16cm0106520h0001 and 18cm0106535h0001); by Grants-in-Aid for Practical Research for Innovative Cancer Control from AMED (16ck0106089h003 and 18cm0106535h0001); by National Cancer Center Research and Development Funds (26-A-1 and 26-A-3), and by the Takeda Science Foundation. The super-computing resource was provided by Human Genome Center, the Institute of Medical Science, the University of Tokyo.

## Author contributions

A.K., A.Y., E.K., H.F., H.H., H.K., M.K., N.N., S. Iwata, T.G., T.H., Y.I., and Y.N. provided samples and clinical data. H.I., M.H., M.S., N.A., S. Mitani, and Y.T. performed sample acquisition and processing for sequence. A.Y., D.M., T.M., and Y.O. performed pathological review. H.I., H.N., M.F., M.S., N.A., S. Mitani, and T.S. performed sequence analysis. K.K., R.Y., S. Imoto, and S. Miyano coordinated data acquisition from TCGA. H.N., K.K., M.F., R.Y., S. Imoto, and S. Miyano performed bioinformatics analyses. H.I., K.K., K.M., M.H., N.A., and R.Y. directed the research and wrote the paper, with contributions from A.K., H.N., M.F., R.N., S. Imoto, S. Miyano, T.K., and T.M.

## Competing interests

The authors declare no competing interests.

## Additional information

Makoto Hirata [1,28], Naofumi Asano[2,3,28], Kotoe Katayama[4,28], Akihiko Yoshida [5], Yusuke Tsuda[1,6], Masaya Sekimizu[7], Sachiyo Mitani[7], Eisuke Kobayashi[8], Motokiyo Komiyama[9], Hiroyuki Fujimoto[9], Takahiro Goto[10], Yukihide Iwamoto[11,12], Norifumi Naka[13], Shintaro Iwata[8,14], Yoshihiro Nishida[15], Toru Hiruma[16], Hiroaki Hiraga[17], Hirotaka Kawano[6,18], Toru Motoi[19], Yoshinao Oda[20], Daisuke Matsubara[21], Masashi Fujita [22], Tatsuhiro Shibata[23], Hidewaki Nakagawa[22], Robert Nakayama[3], Tadashi Kondo[2], Seiya Imoto [24], Satoru Miyano[25], Akira Kawai[8], Rui Yamaguchi[25,29], Hitoshi Ichikawa [7,26,29] & Koichi Matsuda [27,29]

[1]Laboratory of Genome Technology, Institute of Medical Science, University of Tokyo, Tokyo 108-8639, Japan. [2]Division of Rare Cancer Research, National Cancer Center Research Institute, Tokyo 104-0045, Japan. [3]Department of Orthopaedic Surgery, Keio University School of Medicine, Tokyo 160-8582, Japan. [4]Laboratory of Sequence Analysis, Institute of Medical Science, University of Tokyo, Tokyo 108-8639, Japan. [5]Department of Pathology and Clinical Laboratory, National Cancer Center Hospital, Tokyo 104-0045, Japan. [6]Department of Orthopaedic Surgery, Faculty of Medicine, University of Tokyo, Tokyo 113-8654, Japan. [7]Department of Clinical Genomics, National Cancer Center Research Institute, Tokyo 104-0045, Japan. [8]Department of Musculoskeletal Oncology, National Cancer Center Hospital, Tokyo 104-0045, Japan. [9]Department of Urology, National Cancer Center Hospital, Tokyo 104-0045, Japan. [10]Department of Orthopaedic Surgery and Musculoskeletal Oncology, Tokyo Metropolitan Cancer and Infectious Diseases Center Komagome Hospital, Tokyo 113-8677, Japan. [11]Department of Orthopaedic Surgery, Graduate School of Medical Sciences, Kyushu University, Fukuoka 812-8582, Japan. [12]Kyushu Rosai Hospital, Kitakyushu 800-0296, Japan. [13]Musculoskeletal Oncology Service, Osaka International Cancer Institute, Osaka 541-8567, Japan. [14]Division of Orthopaedic Surgery, Chiba Cancer Center, Chiba 260-8717, Japan. [15]Department of Rehabilitation Medicine, Nagoya University Graduate School of Medicine, Nagoya 464-8601, Japan. [16]Department of Orthopaedic Surgery, Kanagawa Cancer Center, Yokohama 241-8515, Japan. [17]Department of Musculoskeletal Oncology, National Hospital Organization Hokkaido Cancer Center, Sapporo 003-0804, Japan. [18]Department of Orthopaedic Surgery, Teikyo University School of Medicine, Tokyo 173-8606, Japan. [19]Department of Pathology, Tokyo Metropolitan Cancer and Infectious Diseases Center Komagome Hospital, Tokyo 113-8677, Japan. [20]Department of Anatomic Pathology, Graduate School of Medical Sciences, Kyushu University, Fukuoka 812-8582, Japan. [21]Division of Integrative Pathology, Jichi Medical University, Shimotsuke 329-0498, Japan. [22]Laboratory for Cancer Genomics, RIKEN Center for Integrative Medical Sciences, Yokohama 230-0045, Japan. [23]Laboratory of Molecular Medicine, Institiue of Medical Science, University of Tokyo, Tokyo 108-8639, Japan. [24]Division of Health Medical Data Science, Institute of Medical Science, University of Tokyo, Tokyo 108-8639, Japan. [25]Laboratory of DNA Information Analysis, Institute of Medical Science, University of Tokyo, Tokyo 108-8639, Japan. [26]Division of Translational Genomics, National Cancer Center-Exploratory Oncology Research & Clinical Trial Center, Tokyo 104-0045, Japan. [27]Laboratory of Clinical Genome Sequencing, Graduate School of Frontier Sciences, University of Tokyo, Tokyo 108-8639, Japan. [28]These authors contributed equally: Makoto Hirata, Naofumi Asano, Kotoe Katayama. [29]These authors jointly supervised this work: Rui Yamaguchi, Hitoshi Ichikawa, Koichi Matsuda. *email: ruiy@ims.u-tokyo.ac.jp; hichikaw@ncc.go.jp; kmatsuda@edu.k.u-tokyo.ac.jp

