## [Peer Review File · Nature Communications]

Reviewers' comments:

Reviewer #1 (Remarks to the Author):

Hirata describe the somatic genetic copy number and mutational analyses of dedifferentiated liposarcomas (DDLPS) using exome sequencing and RNA sequencing. They sequence 66 new cases and analysed publically available data from 53 TCGA cases. Also, eight pairs of DDLPS plus well-differentiated components (WDLPS) were assessed.

Overall, this presents a thorough analyses of genetic aberrations in a large number of DDLPS. This data is of likely interest and use to the specialist community and the availability of data should be confirmed prior publication, the currently the link seems incomplete.

Some findings such as the gain 12q15 and Jun amplification linked to poor outcome are consistent with the previous studies referenced and not novel. The 2 main clusters described are based on these genetic genetic features (12q15 and 1p32 gain). A small number of histologically verified cases did not have this aberration that is frequently used as a diagnostic adjunct. Are there any unusual features in these cases? Are these DDLPS? Other genetic changes seem distributed across these clusters although data for fusions with DNMS3OS are not available for all.

The finding of CTDSP1/2-DNMS3OS fusion genes seems the most novel finding but is not mentioned in the abstract etc Further details on its expression and role would enhance the paper.

The number of paired WD versus DD LPS components in samples is small and difficult to draw strong conclusions but shows that 1q and 12q gain are in common to WD and DD similar to ref 31 discussed but different to other reports quoted. However the current study does indicate that DD additionally show mutational events that are well-summarised in the model in suppl figure 12 – show in main text?

Reviewer #2 (Remarks to the Author):

I would like to congratulate the Japan Sarcoma Genome Consortium (JSGC) on the completion of the manuscript titled "Integrated Exome and RNA Sequencing of Dedifferentiated Liposarcoma". In this paper, the authors describe the genomic landscape of liposarcoma using data derived from whole exome sequencing of 119 dedifferentiated liposarcomas, of which 53 were from the TCGA. In addition, RNAseq data was available for structural variant and integrated analyses. An exploratory analysis of 8 paired WD/DDLPS samples was performed as well to examine for shared and unique alterations.

Overall, this represents one of the largest genomic landscaping studies of liposarcoma, and contributes to information that had been presented before in several smaller studies. Notably, this study also attempts to correlate genomic alterations to clinical outcome.

I have some comments and suggestions.

1) One of the highlights of this paper is the potential to identify prognostically-relevant genomic parameters. Therefore, the robustness of the clinical information collected is important. Firstly, were only non-metastatic cases included in this study? Secondly, in the analysis of genomic clusters and prognosis, margins and primary site were selected in the multivariate analysis. Why were these parameters specifically included, and other variables such as tumor size and patient demographics not included? I note that Table 2 is incomplete - only the univariate results were shown. Most importantly, what new prognostic information does this paper add that we do not

already know from the published TCGA paper (Cell 2017)?

2) The authors should suggest how their findings can be translated into a clinical setting. There are already good clinical models such as the MSKCC model, Sarculator etc. that can be used to prognosticate liposarcoma. In particular, were there any histological variations observed that correlated with certain SCNAs, distinct clusters, or with the DNM3OS fusions? Can we use a representative gene or set of genes to recapitulate each prognostic cluster?

3) The identification of novel DNM3OS fusions in a subset of liposarcoma represents is interesting. I note the authors claim that this group of tumors are more “proliferative”, presumably based on their correlation to cell cycle gene pathways. However, this is probably an inferred correlation and should be stated in such a way, unless the authors do manage to correlate this group of tumors with a proliferative clinical/in vitro phenotype and perhaps poor outcome. Also, are DNM3OS fusions also found in WDLPS, or are they specific to DDLPS? In vitro functional assessment of this fusion protein will definitely be interesting to perform/follow-up.

4) The authors conclude that “This large scale integrated genomic analysis reveals the mechanisms underlying the development and progression of DDLPS and provides novel insights that could be used to refine DDLPS therapy...” but the large part of the paper is not about discovery or validation of treatment targets. That this result can be used to refine DDLPS therapy is beyond speculation.

5) Other minor comments are below:

- It is unclear to me exactly how many samples were used for RNA sequencing from the methodology section.
- Is there information of the use of chemotherapy, in any form – neoadjuvant/adjuvant/intra-operative, in these patients?
- Referring to lines 174-178, can the authors clarify if 11 or 12 SCNA regions were predictors of PFS?
- In Figure 3B and 3C, labels on the X-axis are required for clarity.

Point-by-point response to Reviewers' comments:

Reviewer #1 (Remarks to the Author):

- Overall, this presents a thorough analyses of genetic aberrations in a large number of DDLPS. This data is of likely interest and use to the specialist community and the availability of data should be confirmed prior publication, the currently the link seems incomplete.

We agree that the current next-generation sequencing data should be available for all researchers. Sequencing FASTQ data files have been deposited at the Japanese Genotype-phenotype Archive (JGA, <http://trace.ddbj.nig.ac.jp/jga>), which is hosted by the DDBJ, under accession number JGAS00000000177 and the data are currently ready for publication after acquisition of the paper approval.

- Some findings such as the gain 12q15 and Jun amplification linked to poor outcome are consistent with the previous studies referenced and not novel.
The 2 main clusters described are based on these genetic genetic features (12q15 and 1p32 gain). A small number of histologically verified cases did not have this aberration that is frequently used as a diagnostic adjunct.
Are there any unusual features in these cases? Are these DDLPS? Other genetic changes seem distributed across these clusters although data for fusions with DNMS3OS are not available for all.

Indeed, this point is very important. Cluster 3 cases from TCGA were excluded, because we could not confirm their histological features in other ways than histological reports. In addition, one of Cluster 3 cases from JSGC was also excluded, because MDM2 or CDK4 amplification or expression was not found by FISH or IHC and also the histological and genomic feature was similar with atypical spindle cell lipomatous tumor (ASLT). Other five cases in Cluster 3 from JSGC remain included for the current study, because their histological features were compatible with WHO classification of DDLPS though some of them actually exhibited weak or sparse MDM2 expression by immunohistochemistry.

We also probed the common genomic features of Cluster 3 and found that copy-number of 157 genes in nine regions were consistently gained or lost in the five case. Further pathway analysis found some of the genes were associated with PI3K-AKT signaling pathway. *TP53* mutations were also accumulated in Cluster 3 (Fig. 3a). Hence, *TP53* mutation and/or PI3K-AKT signaling pathway may play a pivotal role in the development of DDLPS with Cluster3.

We added Supplementary Table 7, describing the genes commonly gained or lost in Cluster 3 tumors (p.13, line 206-209).

- The finding of CTDSP1/2-DNMS3OS fusion genes seems the most novel finding but is not mentioned in the abstract etc Further details on its expression and role would enhance the paper.

The finding of the novel fusion genes is one of the highlights of the current study. We added the finding of the fusion genes in the abstract. In addition, we examined if the fusion genes were also found in WDLPS samples (not the WD components of DDLPS). None of the 17 WDLPS samples harbored the novel fusion genes, suggesting that the novel fusion genes might be involved with malignant transformation of DDLPS. We added the results in p.11, line 173-176.

- The number of paired WD versus DD LPS components in samples is small and difficult to draw strong conclusions but shows that 1q and 12q gain are in common to WD and DD similar to ref 31 discussed but different to other reports quoted. However, the current study does indicate that DD additionally show mutational events that are well-summarised in the model in suppl figure 12 – show in main text?

Thank you for your suggestion. We changed Supplementary Figure 12 to Figure 5.

Reviewer #2 (Remarks to the Author)

- 1) One of the highlights of this paper is the potential to identify prognostically-relevant genomic parameters. Therefore, the robustness of the clinical information collected is important. Firstly, were only non-metastatic cases included in this study? Secondly, in the analysis of genomic clusters and prognosis, margins and primary site were selected in the multivariate analysis. Why were these parameters specifically included, and other variables such as tumor size and patient demographics not included? I note that Table 2 is incomplete - only the univariate results were shown. Most importantly, what new prognostic information does this paper add that we do not already know from the published TCGA paper (Cell 2017)?

We checked the status of the patients, but none of them had the evidence of metastasis at diagnosis.

We also examined other clinical parameters but no other parameters were shown to be associated with clinical prognosis by univariate analysis. As multivariate analysis results in Table 2 somehow dropped off in the first submission, the multivariate results were restored in the revised table. We apologize for our fault.

The TCGA paper included the methylation status for the clustering, which may be difficult for clinical application because of increased cost and time. In contrast, the current study identified gain of 1p32.1 as a sole independent predictor for poor disease-specific survival. As copy-number status of 1p32.1 can be easily examined by FISH or other conventional methods, we believe that this genomic classification is more expedient for clinical application.

- 2) The authors should suggest how their findings can be translated into a clinical setting. There are already good clinical models such as the MSKCC model, Sarculator etc. that can be used to prognosticate liposarcoma. In particular, were there any histological variations observed that correlated with certain SCNAs, distinct clusters, or with the DNM3OS fusions? Can we use a representative gene or set of genes to recapitulate each prognostic cluster?

It was hard to apply the current cases to MSKCC or other models, as some essential clinical parameters for the model were missing in the current cohort. But indeed we believe the models can predict clinical outcome more precisely by including our genomic clustering criteria. We discussed these points in the Discussion in p. 20, line 328-331.

Histological features were also re-examined according to the genomic clustering or fusion status, but most of them exhibited similar morphology and difficult to be differentiated histologically. However, the genes, whose expression level was dependently modulated on prognostic SCNAs (Supplementary Table 10), can be the potent prognostic marker, though further validation analysis, using other cohort samples, is required for the clinical application.

- 3) The identification of novel DNM3OS fusions in a subset of liposarcoma represents is interesting. I note the authors claim that this group of tumors are more “proliferative”, presumably based on their correlation to cell cycle gene pathways. However, this is probably an inferred correlation and should be stated in such a way, unless the authors do

manage to correlate this group of tumors with a proliferative clinical/in vitro phenotype and perhaps poor outcome. Also, are DN3OS fusions also found in WDLPS, or are they specific to DDLPS? In vitro functional assessment of this fusion protein will definitely be interesting to perform/follow-up.

We revised the expression of this point as follows.

p. 11, line 166-167: “indicating the association of DN3OS fusion genes with cell-cycle modulation in DDLPS.”

p. 19, line 308-310: “DDLPS that contained DN3OS fusion genes showed the significant upregulation of DN3OS and were correlated to cell-cycle pathway as compared with those without fusion genes.”

- 4) The authors conclude that “This large scale integrated genomic analysis reveals the mechanisms underlying the development and progression of DDLPS and provides novel insights that could be used to refine DDLPS therapy...”but the large part of the paper is not about discovery or validation of treatment targets. That this result can be used to refine DDLPS therapy is beyond speculation.

We revised the final sentence in Abstract as follows.

p. 5, line 75: “... and provides novel insights that could contribute to refinement of DDLPS management.”

- 5) Other minor comments are below:
 - It is unclear to me exactly how many samples were used for RNA sequencing from the methodology section.

The number of the samples used for RNA sequencing was added in Materials and Methods (p. 26, line 430 and 438).

- Is there information of the use of chemotherapy, in any form
- neoadjuvant/adjuvant/intra-operative, in these patients?

The report from TCGA did not contain the information of chemotherapy, either. As the information of chemotherapy was not available in most cases, this parameter was not included in the current study.

It may be because systemic chemotherapy might not be performed in most cases, as DDLPS is resistant to chemotherapy.

- Referring to lines 174-178, can the authors clarify if 11 or 12 SCNA regions were predictors of PFS?

We also performed multivariate Cox regression analysis for PFS, including clinical parameters and three regions, independently associated with PFS; the high-level gain of 4p16.3 and 6p21.1, and the loss of 9q34.11, and found that these three as well as primary tumor site were independent predictors for poor PFS. Hence, we added Supplementary Table 8.

- In Figure 3B and 3C, labels on the X-axis are required for clarity.

We added the label in Figure 3B and 3C. Thank you.

Thanking all the reviewers and the Editor in advance for helping us to ameliorate our manuscript and the robustness of our results, we hope that the paper is now suitable for publication in Nature Communications.

Reviewers' comments:

Reviewer #1 (Remarks to the Author):

Re Hirata et al v2 . The large data cohort, identification of a novel fusion gene and clinical correlates still stand out as of interest. I have the following comments regarding their rebuttal:

"we examined if the fusion genes were also found in WDLPS samples (not the WD components of DDLPS). None of the 17 WDLPS samples harbored the novel fusion genes," does this mean that the WD components were not tested or that the WD components did not have the fusion. This should be clarified or tested to support or otherwise the suggestion that the novel fusion is involved in dedifferentiation.

"It was hard to apply the current cases to MSKCC or other models, as some essential clinical parameters for the model were missing in the current cohort. But indeed we believe the models can predict clinical outcome more precisely by including our genomic clustering criteria." evidence is not very clearly supporting this point

"the genes, whose expression level was dependently modulated on prognostic SCNAs (Supplementary Table 10), can be the potent prognostic marker, though further validation analysis, using other cohort samples, is required for the clinical application." This point is not clearly made in the revised text.

Both reviewers 1 and 2 indicate that determining a mechanistic role of the fusion protein could support the assumed functions presented from their pathway analyses and would be of interest and enhance the study. I presume that the authors unfortunately think that this is beyond their current scope?

Other points have been adequately addressed.

Reviewer #2 (Remarks to the Author):

I would again like to congratulate the Japan Sarcoma Genome Consortium (JSGC) on the completion of the manuscript titled "Integrated Exome and RNA Sequencing of Dedifferentiated Liposarcoma".

The reply from the authors is adequate and they have made significant improvements to the manuscript. I have no further comments for them and would recommend publication of their work.

Point-by-point response to Reviewers' comments:

Again, we would like to express our deepest gratitude for the all the valuable and constructive comments from all the reviewers and the Editor. According to the comments, we revised the manuscript as mentioned below.

Note: All changes in text, tables, and figures were highlighted in red.

Reviewer #1 (Remarks to the Author):

- “we examined if the fusion genes were also found in WDLPS samples (not the WD components of DDLPS). None of the 17 WDLPS samples harbored the novel fusion genes,”

Does this mean that the WD components were not tested or that the WD components did not have the fusion?

This should be clarified or tested to support or otherwise the suggestion that the novel fusion is involved in dedifferentiation.

We also examined the fusion status of the eight WD components, none of which harbored DN3OS fusions. We describe the result in the main text (p.11, line 171-172).

- “It was hard to apply the current cases to MSKCC or other models, as some essential clinical parameters for the model were missing in the current cohort. But indeed we believe the models can predict clinical outcome more precisely by including our genomic clustering criteria.”

Evidence is not very clearly supporting this point.

Multivariate analysis including clinical parameters and genomic clustering indicated the genomic clustering as an independent predictor of prognosis (Table 2). Also, further multivariate analysis using the genomic clustering and clinical parameters, which mostly compose MSKCC nomogram, also showed the independency of the genomic clustering (p.20, line 325-329). We believe establishment of novel nomogram including the genomic clustering is beyond the purpose of the current study, but these results of multivariate analysis could support the possibility to contribute to more precise prediction of the nomogram by our genomic clustering.

- “the genes, whose expression level was dependently modulated on prognostic SCNAs (Supplementary Table 10), can be the potent prognostic marker, though further validation analysis, using other cohort samples, is required for the clinical application.” This point is not clearly made in the revised text.

Further survival analysis after stratifying TCGA cohort by the gene expression level, showed that some of the genes are associated with clinical prognosis (Supplementary Figure 9) (on page 15, line 240-241). We also added discussion on this point in the text (page 21, line 335-338).

(We mislabeled the supplementary table number; the table of the genes at prognostic SCNAs was Supplementary Table 11.

We apologize for having caused any confusion.)

- Both reviewers 1 and 2 indicate that determining a mechanistic role of the fusion protein could support the assumed functions presented from their pathway analyses and would be of interest and enhance the study. I presume that the authors unfortunately think that this is beyond their current scope?

We agree that the function of the novel fusion gene is one of the curious points in this study and further analysis may bring out important clinical implications. However, we think this point is beyond the current scope as mentioned on page 19, line 303-304.

Thanking all the reviewers and the Editor in advance for helping us to ameliorate our manuscript and the robustness of our results, we hope that the paper is now suitable for publication in Nature Communications.

Your Sincerely

Koichi Matsuda

REVIEWERS' COMMENTS:

Reviewer #1 (Remarks to the Author):

I think that the authors have adequately addressed and clarified the points raised in the revised version of their manuscript.